# Analysis of Expression of the ANG1, CaSR and FAK Proteins in Uterine Fibroids

**DOI:** 10.3390/ijms25137164

**Published:** 2024-06-28

**Authors:** Anna Markowska, Mateusz de Mezer, Paweł Kurzawa, Wiesława Bednarek, Anna Gryboś, Monika Krzyżaniak, Janina Markowska, Marian Gryboś, Jakub Żurawski

**Affiliations:** 1Department of Perinatology and Women’s Diseases, Poznan University of Medical Sciences, 60-535 Poznan, Poland; amarkowska@ump.edu.pl; 2Medical Biology, Department of Immunobiology, Poznan University of Medical Sciences, 60-806 Poznan, Poland; mdemezer@ump.edu.pl; 3Department of Clinical Pathology and Immunology, Poznan University of Medical Sciences, 60-355 Poznan, Poland; pawel.kurzawa@skpp.edu.pl (P.K.); monika.krzyzaniak@skpp.edu.pl (M.K.); 4Department of Oncological Pathology, University Clinical Hospital in Poznan, Poznan University of Medical Sciences, 61-848 Poznan, Poland; 5Department of Oncological Gynecology and Gynecology, Medical University of Lublin, 20-059 Lublin, Poland; wbed@wp.pl; 6Department of Gynecology and Obstetrics, Faculty of Health Sciences, Wroclaw Medical University, 50-367 Wroclaw, Poland; annagrybos@yahoo.pl; 7Gynecological Center, 60-560 Poznan, Poland; jmarkmed@poczta.onet.pl; 8Institute of Health Sciences, University of Opole, 45-040 Opole, Poland; margrybos@gmail.com

**Keywords:** uterine myoma, angiopoietin 1, calcium sensing receptor, focal adhesion kinase

## Abstract

Understanding the molecular factors involved in the development of uterine myomas may result in the use of pharmacological drugs instead of aggressive surgical treatment. ANG1, CaSR, and FAK were examined in myoma and peripheral tissue samples taken from women after myoma surgery and in normal uterine muscle tissue samples taken from the control group. Tests were performed using tissue microarray immunohistochemistry. No statistically significant differences in ANG1 expression between the tissue of the myoma, the periphery, and the normal uterine muscle tissue of the control group were recorded. The CaSR value was reduced in the myoma and peripheral tissue and normal in the group of women without myomas. FAK expression was also lower in the myoma and periphery compared to the healthy uterine myometrium. Calcium supplementation could have an effect on stopping the growth of myomas.

## 1. Introduction

Uterine myomas are benign monoclonal tumors originating from smooth muscle cells and affect more than 70% of women of reproductive age [1]. Clinical symptoms affect 20–50% of these cases and include excessive uterine bleeding, pelvic pressure and pain, bowel and bladder dysfunction, impaired fertility, and complications during pregnancy and delivery. The symptoms experienced are related to the size and location of the myoma within the uterus (subserous, intramural, or submucous) [2,3].

The development and growth of myomas depend on many factors, including steroid hormones (estrogen and progesterone), numerous cytokines (including TNF-α) and growth factors (TGF-β, activin A, and PDGF), the extracellular matrix (ECM), microRNA, genetic factors, stem cells, and vitamin D deficiency [2,4,5,6].

Estrogen receptors α and β influence the production of numerous cytokines and growth factors, both those produced in the ovary and those resulting from the conversion of ovarian and adrenal androgens. This makes the myoma susceptible to the effects of progesterone, which is involved in regulating genes responsible for proliferation and apoptosis. In addition, both estrogen and progesterone are involved in the production and accumulation of ECM by increasing the expression of fibronectin and collagen [2,4]. A small non-coding microRNA has also been found to be involved in myoma development, mainly affecting ECM formation. The overexpression of miR-21 has been shown to play a role in the pathomechanism of uterine myoma development. Through the TGF-β3 pathway, myoma cells showed increased expression of metalloproteinases (MMP2, MMP9, MMP11) and Serpin1 [5].

A study by Firdaus et al. [6] showed that 40–50% of tumors present chromosomal aberrations in exon *2MED12*.

According to Islam et al. [2] and Santamaria et al. [7], stem cells located in the myometrial layer are involved in the development of myomas. Their activity occurs through the Hippo pathway associated with proliferation and apoptosis and the canonical WNT/β catenin pathway.

The results of numerous studies also indicate the role of vitamin D deficiency in the development of myomas. Data support the significance of vitamin D supplementation in reducing uterine fibroid growth [8,9].

Although the molecular mechanisms of the development and growth of myomas are only partially understood, they still represent an interesting model for research related to the conservative treatment of myomas. Numerous attempts at pharmacological management, including the use of hormones, e.g., GnRH agonists, GnRH antagonists, and SPRMS (selective progesterone receptor modulators), have so far proven unsatisfactory [3,10,11]. Therefore, the search for the activity of new molecules—which, as in malignant tumors, may be involved in inhibiting the development of myomas—is ongoing. These molecules may include factors affecting angiogenesis, cell adhesion, or ECM [12,13,14].

Angiogenesis is a complex process involving stimulatory compounds, ECM components, and various cell types. It plays an important role in abnormal vasculature and fibroid growth [15,16,17]. Angiogenesis-stimulating factors include angiopoietins.

Angiopoietin 1 (ANG1) is one of the three best-known angiopoietins encoded by *ANGPT1*, a gene located on chromosome 8q23.1. It is a ligand of the receptor with tyrosine kinase 2 (Tie-2) activity. It causes the remodeling of immature vessels and stabilizes mature vessels [17]. It is expressed on smooth muscle cells, fibroblasts, and pericytes and induces the adhesion, migration, and survival of endothelial cells. ANG1 synthesis is regulated by the hypoxia-inducible factor HIF-1α and is associated with calcium ions. Angiopoietin 1 exerts a local effect on the ECM. Cell signal transduction is mediated by protein kinases, including MAPK and FAK [2,18,19].

Angiogenic functions of ANG1 and ANG2 have been shown to be modulated by Ca^2+^-dependent calcium signaling pathways [16,19].

Endothelium-derived Ca^2+^ enhances angiogenesis through multiple factors, including the aforementioned ANG1 and fibroblast growth factor [19]. One mechanism promoting angiogenesis may be the influx of Ca^2+^ ions into the mitochondria, resulting in hypoxia associated with activating proangiogenic factors [16]. By contrast, a study by Yang et al. [20] showed that under hypoxic conditions, Ca^2+^ concentrations were higher in myoma than in normal uterine muscle cells, and this was responsible for the susceptibility of myomas to apoptosis. Changes in Ca^2+^ concentrations in myoma were also dependent on estrogen concentrations [21].

The final stage of myoma degeneration is the calcification of its cells, which is observed as a therapeutic effect of embolization [22]. Calcium hemostasis is regulated by CaSR, the expression of which has been found mainly in the parathyroid glands, kidneys, and brain. It is involved in many biological processes of the body. It was recently discovered that CaSR regulates the fate of many cells in the body; its increased or decreased expression has been described in various types of cancer [23]. Its presence in uterine myomas has not yet been described.

Focal adhesion kinase (FAK) is a non-receptor tyrosine kinase, the product of the 8q24-localized *PTK2* gene. It is found on cells that form connections with ECM or other cells—these are foci of adhesion. The primary function of FAK is to transmit signals from the external environment through receptors to the inside of the cell, but also in the opposite direction. Its receptors can be integrins or growth factors. In addition to its involvement in endothelial cell migration and survival, FAK has also been found to be closely involved in controlling angiogenesis [24,25]. Its action is mainly through the ERK1/AKT and JNK signaling pathways [26,27]. According to Tavora et al. [28], FAK is a possible target for anti-angiogenic targeted therapies in malignant neoplasms.

Although FAK1 is often studied in various types of endometrial cancers and is attributed to a role in regulating cell growth [29], the level of this protein in benign lesions such as uterine fibroids has not been investigated to date. Of the various attempts at conservative treatment of uterine fibroids, not all of them have been accepted (e.g., the hormonal drug ulipristal acetate was withdrawn). The clinician and histopathologist believe that the deposition of calcium salts in the myoma cells is related to the inhibition of its growth (this is proven by the results of embolization surgical procedure). This process involves Ang 1, which promotes vascular integrity and is involved in calcium signaling pathways. However, FAK, in addition to controlling the angiogenesis process through cell adhesion (including the fibroid), prevents its growth. Thus, the action of the three molecules we studied seems closely related. The study aimed to determine the importance of ANG1, CaSR, and FAK in uterine fibroids that could be applied in targeted therapy.

## 2. Results

No significant differences between the tumor center, tumor periphery, and controls for ANG1 (*p* = 0.983) were recorded (Figure 1A). The average level of ANG1 was from 22,355.37 ± 9961.68 for the tumor center to 22,900.08 ± 10,762.30 for the controls (Table 1 and Figure 2A).

CaSR1 expression was significantly different between the samples (*p* = 0.001) (Figure 1B). Post hoc analysis confirmed that it was lower in the tumor center (median = 34.31) and tumor periphery (median = 14.00) than in the controls (median = 1056.15) (Table 1 and Figure 2B).

A similar result was confirmed for FAK: expression was significantly different between the samples (*p* < 0.001) (Figure 1C) and post hoc analysis confirmed that expression was lower in the tumor center (16,037.69 ± 11,342.30) and tumor periphery (18,338.54 ± 8824.04) than in the controls (39,665.24 ± 14,231.92) (Table 1 and Figure 2C).

## 3. Discussion

Uterine myomas are heterogeneous tumors by number, size, location, and vascularization. Their numerous associated symptoms—such as bleeding, anemia, pressure on the bladder, and fertility disorders—seriously affect women’s health [4,30,31]. Numerous factors and molecular signaling pathways have been studied for many years, suggesting therapeutic alternatives to surgery [3,5,8,32].

Studies have shown vascularization’s critical effect on myomas’ development [17,18,30]. Angiopoietin 1 (ANG1) expression on smooth muscle and perivascular cells plays an essential role in vascular maturation and mediates cell migration, adhesion, and survival [17,19,33].

In the study presented here, ANG1 showed expression in both myoma tissue and surrounding myoma tissue, as well as in the unaltered uterine myoma of women in the control group. However, in the 70 women in this study (both myoma and peripheral tissue) as well as the 12 women in the control group (healthy uterine muscle), the values were not significantly different. In the study by Nakayama et al. [34], ANG1 expression in myoma tissue was examined in 17 women, and 13 (76.5%) were found to be positive for ANG1 expression. However, ANG1 values were not determined in the surrounding tissue or in the control group. We are the first to determine the ANG1 expression of myomas, as well as in the surrounding tissue and in the unaltered uterine muscle.

The important role of angiopoietin 1 is evidenced by papers on its effect on endometrial bleeding and its beneficial effect in reducing the risk of early cervical cancer metastasizing to lymph nodes [35,36]. According to studies on angiogenic mechanisms, fibroid vascularization requires further investigation in order to understand myoma development pathways [30].

Angiopoietins (including ANG1) regulate angiogenesis and vasculogenesis through a variety of pro-angiogenic, anti-angiogenic, and growth factors, but also through calcium (Ca^2+^)-related signaling [16,19].

The serum Ca^2+^ levels in 194 women in Nigeria, among whom 97 had uterine myomas and 97 formed the control group, have been described. The group with uterine myomas were found to have low Ca^2+^ levels, and dietary supplementation with this element was suggested [37]. Controversial data come from studies on Chinese women. In a case–control study, Ca^2+^ concentrations were not statistically different (267 and 267 women, respectively) between them [38]. Kopf et al. [39] described a case of a large myoma in a pregnant woman (weighing 2834 g) that caused hypercalcemia with significant morbid symptoms that resolved after tumor removal.

The above-described controversial results of serum Ca^2+^ determinations were the reason for the determination of the calcium receptor (CaSR) in the myometrial tumor, its periphery, and the normal muscle of control women. Such determinations have not been encountered in the available literature. In 70 women with uterine myomas, CaSR was determined in the tumor and its periphery. In both the myoma tumor and its periphery, CaSR concentrations were lower than in the myometrium of healthy women. The values of these differences were statistically significant. This result supports the suggestion by researchers at the University Hospital of Nigeria that calcium supplementation could prevent the development of large myomas in women with small myomas.

The relationship between angiogenesis, FAK1 expression, and Ca^2+^ concentration has been described [19,40]. ANG1 can induce FAK phosphorylation, but at the same time, ANG1 is associated with calcium activity [19]. This was the rationale for studies on the mentioned molecules. There are no studies on FAK in uterine myomas in the literature.

In the pathology of many tumors, especially malignant ones, FAK is involved in the transmission of signals to and from the cell, migration, cell survival, drug resistance, and tumor invasion participating in its progression. Clinical trials with FAK inhibitors, mainly in combination with chemotherapy, are currently underway [40,41,42].

## 4. Materials and Methods

### 4.1. Study Design

The study group consisted of tissue material from 70 patients with uterine myomas diagnosed through histopathological examinations performed by two independent pathologists. The selected women ranged in age from 24 to 82, with a mean age of 50. The myomas ranged in diameter from 1 cm to 10 cm, with an average of 4.3 cm in the largest dimension. FIGO classification was not used as the material originated from different clinics, but no pedunculated fibroid was found in the uterine cavity or the cervix. They were mainly FIGO grade 3 to 6 fibroids, according to Munro et al., 2011 [43].

The control group (Appendix A) included tissue sections, morphologically unchanged, from 12 women without known myomas after histopathological examination. These women had undergone surgery for endometrial hypertrophy (hyperplasia endometrii) or uterine prolapse. They were between 56 and 69 years old, with a mean age of 61 years.

The patients included in our study—both those with myomas and those in the control group—had no chronic disease. They were selected for the myoma study group and the control group for abnormal uterine bleeding (endometrial hyperplasia) and genital prolapse (genital prolapsus). They were qualified for surgery by an internist and an anesthesiologist. No diseases considered to be risk factors were identified in the participants, such as in the case of endometrial cancer (obesity). The patients took no hormones for at least a month before the fibroid removal procedure. Therefore, a table containing clinical data was not included, so no correlation was made with the expression of the proteins analyzed. Thus, only the age of the patients was taken into account.

### 4.2. Immunohistochemical Protein Detection

Protein detection was carried out using the following antibodies: CaSR—calcium sensing receptor antibody (Affinity Biosciences #AF6296); ANG1—angiopoietin 1 antibody (Affinity Biosciences #AF5184); FAK—FAK antibody (Affinity Biosciences #AF6397).

The study was conducted on tissue arranged in tissue microarray blocks (TMAs) prepared according to the procedure described above [9] from placed next to each other 70 uterine myomas and 70 uterine tissues (tumor margin) identified by a pathologist. The material was derived from uterine leiomyoma and muscle tissues outside the uterine myoma in its vicinity of the same patient. Each fragment consisted of elongated smooth muscle cells without atypia, with a very low mitotic rate, i.e., less than one mitosis/10 HPF (high power field). Upon microscopic examination, the tissue sections were found to consist of smooth muscle cells that revealed no atypia and showed no significant morphological changes in the normal myometrium, tumor peripheral tissue, and tumor cells. There was no necrosis or other regressive changes. The only microscopic change was found in the distorted architecture of the tumor tissue. TMAs were assembled using the UNITMA Quick-Ray^®^ Manual Tissue Microarrayer (UNITMA Co., Ltd. Seoul, Republic of Korea). Each TMA contained 14 patient tissue sections and 2 control sections. The tissue cores were 5.0 mm in diameter. The sections intended for histopathological diagnosis preceding the described study were stained with hematoxylin and eosin. Each microarray also contained a fragment of normal uterine tissue from the control group. The control group included tissue slices, morphologically unchanged, from 11 women without known fibroids upon histopathological examination. These women were operated on due to endometrial hyperplasia or genital prolapse. They were aged from 56 to 69 years, averaging 61. The women in the study and control groups had no additional comorbidities. Serial 4-micrometer tissue sections were cut from the donor blocks containing cores of lesions and applied to adhesion slides (Epredia™ SuperFrost Plus™). Slides were stained on a fully automated immunohistochemistry slide stainer, Bench-Mark ULTRA (Ventana Roche, Oro Valley, AZ, USA). The staining protocol parameters were based on HIER using CC1 (a heating time of 24 min, at 100 °C), protease 3 (760-2020) for 4 min, 32 min of incubation with the primary Ab, and OptiView (760-700) with amplification (760-099) as a detection system. The antigen was localized using chromogen DAB-3.3 applied in all the preparations. The slides were stained with hematoxylin II (790-2208) for 8 min and bluing reagent (760-2037) as a post counterstain for 4 min. The slides were passed through a series of alcohols and, finally, xylene before the coverslips were mounted.

For immunohistochemical reactions, tissue material from sections of the normal uterus was used as a positive control. The negative control for immunohistochemical reactions on the same material was subjected to the same procedure without using the original antibody during staining. In addition, using a TMA containing diverse biological material and control tissues allowed for the observation of various immunohistochemical images within each set of tissues. Thus, they provide a control system in relation to each other to evaluate the correctness of the immunohistochemical reactions.

Evaluation of the distinction between the myoma, both the center and periphery of the tumor, from the surrounding normal uterine tissue was performed by an experienced pathologist. In the opinion of the same specialist, no features indicative of necrosis were observed in the analyzed fragments.

### 4.3. Semi-Quantitative Evaluation of CaSR, ANG1, and FAK Protein Expression

For each patient, ten images were taken at a total magnification of 400×. For this purpose, an Olympus Grundium Ocus 40 microscope scanner (Olympus, Tokyo, Japan) was used. Based on the abovementioned photographic documentation, a semi-quantitative evaluation of immunohistochemical reactions was performed in the Olympus commercial cellSens dimensional program. In the program, phase analysis of the stained preparation was performed by automatically detecting objects due to their color (brown chromogen DAB-3.3). Threshold values were entered, according to which the software automatically classified the data. At the preparation stage, the number of cells and the area of immunohistochemical reactions were evaluated. The obtained surface area values were expressed in mm^2^. The measurement results were automatically exported to MS Excel sheets for further statistical analysis [9,44].

### 4.4. Statistical Analysis

The average protein expression level in each sample was calculated as the median from 10 captured pictures. The normality of distribution by subgroup (tumor center, tumor periphery, controls) was verified using the Shapiro–Wilk test and skewness and kurtosis values. Subgroup comparison was carried out with ANOVA (Tukey post hoc test) or the Kruskal–Wallis test (Dunn post hoc test with Bonferroni correction). A significance level of 0.05 was assumed. Analysis was conducted using R 4.0.5. statistical software (R Core Team (2021). R: Language and environment for statistical computing by R Foundation for Statistical Computing, Vienna, Austria).

## 5. Conclusions

In our study, we detected that FAK expression is significantly lower in uterine myomas and their periphery compared to healthy uterine muscle tissue. This fact indicates that FAK is not involved in the growth of benign tumors, and therefore, any clinical trials of FAK inhibitors are out of the question. It may be yet another histological biomarker that distinguishes benign from malignant tumors.

## Figures and Tables

**Figure 1 ijms-25-07164-f001:**
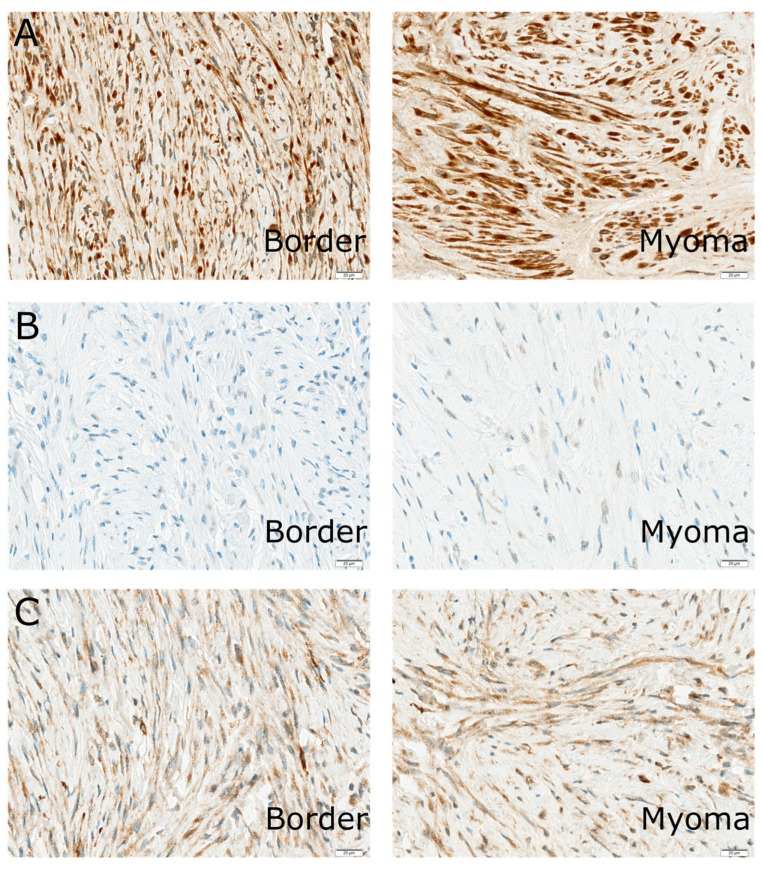
Immunohistochemical expression of the analyzed protein in the uterus of the same patients: ANG1 (**A**), CaSR (**B**), and FAK (**C**). Magnification for all pictures is 400×.

**Figure 2 ijms-25-07164-f002:**
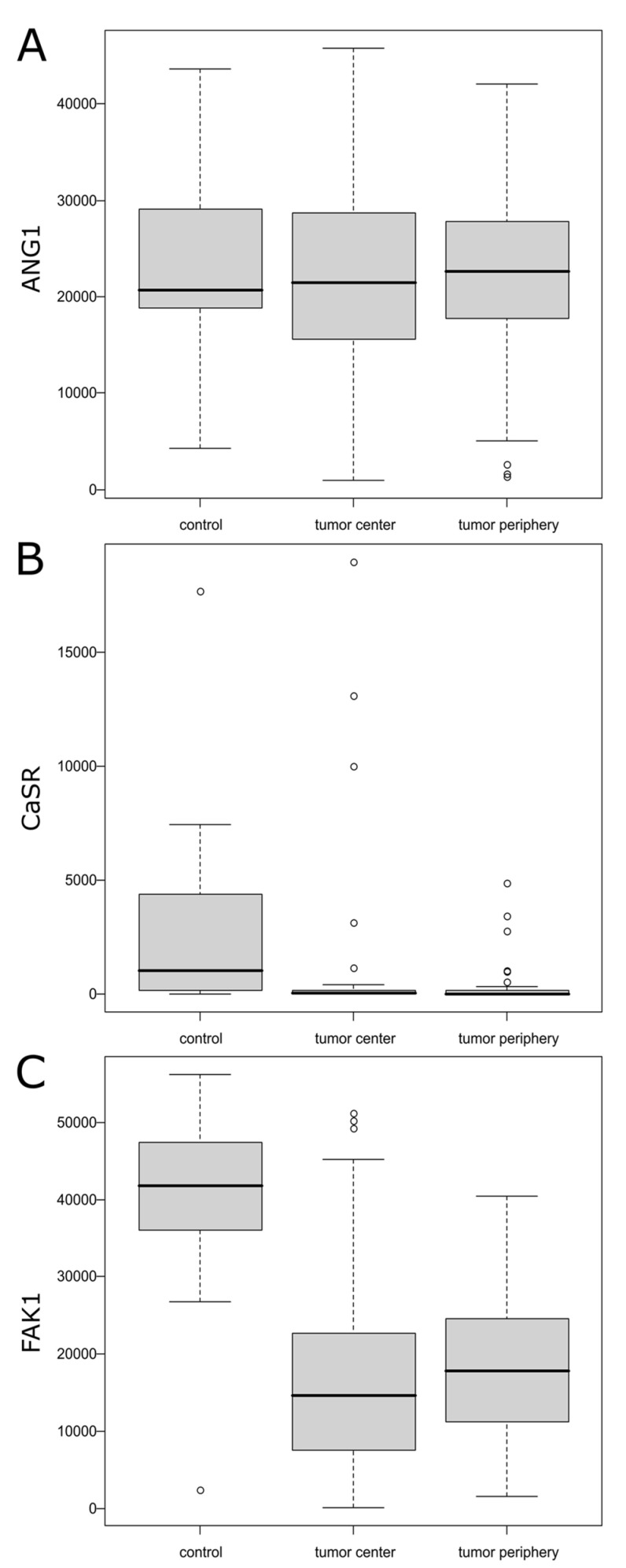
The boxplot for ANG1 (**A**), CaSR (**B**), and FAK (**C**) shows the comparison of expression levels determined by measurement of the intensity of stain resulting from the attachment of the appropriate antibody between the tumor center, the tumor periphery, and the control group.

**Table 1 ijms-25-07164-t001:** Comparison of tumor center and tumor periphery vs. controls for ANG1, CaSR, and FAK.

	N	Mean ± SD/ Median (Q1; Q3)	Range	*p*	Post hoc
ANG1					
Tumor center	70	22,355.37 ± 9961.68	(940.08; 45,829.46)	0.983	
Tumor periphery	70	22,370.11 ± 8540.59	(1313.09; 42,077.36)		
Control	12	22,900.08 ± 10,762.30	(4256.68; 43,689.11)		
**CaSR**					
Tumor center	70	34.31 (7.64; 179.37)	(0.00; 18,997.74)	0.001	Center < control Periphery < control
Tumor periphery	70	14.00 (2.08; 145.02)	(0.00; 4906.41)		
Control	12	1056.15 (171.81; 4343.91)	(1.38; 17,712.65)		
**FAK**					
Tumor center	70	16,037.69 ± 11,342.30	(66.05; 51,188.30)	<0.001	Center < control Periphery < control
Tumor periphery	70	18,338.54 ± 8824.04	(1601.30; 40,503.59)		
Control	12	39,665.24 ± 14,231.92	(2358.86; 56,274.69)		

Groups’ comparison with ANOVA (ANG1, FAK) with the Tukey post hoc test or with the Kruskal–Wallis test (CaSR) with Dunn’s post hoc test including Bonferroni correction for multiple comparisons.

## Data Availability

The data presented in this study are available on request from the corresponding author.

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
