# Peer review of "Analysis of Expression of the ANG1, CaSR and FAK Proteins in Uterine Fibroids"

_ijms, 2024, doi:10.3390/ijms25137164_

Round 1

Reviewer 1 Report

Comments and Suggestions for Authors

This study looks at IHC expression of ANG1, CaSR, and FAK in myoma and peripheral tissue samples taken from women after myoma  surgery and in normal uterine muscle tissue samples taken from the control group. The aim of this study was to identify molecular targets in myomas that could be used for therapy.

Major concerns:

1.     Did the authors take into consideration the role of endogenous hormones on the expression of ANG1, CaSR, and FAK? Are the authors able to sort samples in premenopausal women by menstrual cycle phase (proliferative vs. secretory) to look at regulation of expression of these proteins? Does it change expression patterns?

2.     No history of whether patients where on hormone replacement therapy, or birth control pills were taken or other exogenous forms of estrogen/progesterone at time of sampling and role these exogenous hormones could play on expression of proteins examined.

3.     The activated/phosphorylated forms of these FAK and ANG1 should be assessed in tissues. These are signaling molecules (tyrosine kinases) and expression of total FAK or ANG1 gives no information on whether the protein is activated and has initiated downstream signaling pathways. These studies are critical studies needed for this manuscript.

4.     Please clarify tumor margin. Is it at the periphery of the tumor and consists of tumor tissue? OR is it non-tumor tissue at the periphery of the tumor? It is unclear as currently stated in the methods, although in the results it states, “tumor center, “tumor periphery.”

5.     Please describe in more detail the 14 tissue samples on each TMA. Were patient-matched paired tumor and tumor margin samples placed in the same block and stained under similar conditions? OR were all tumors in one TMA and tumor margin samples in another TMA. Authors only state clearly there are  2 non-tumor control samples per TMA. Please clarify.

6.     Why were non-tumor controls used as positive controls for the different antibodies (Abs)? The authors should have used positive control tissue that is known to highly express the proteins of interest. For example, caSR (human parathyroid or  kidney); for FAK (brain or lung) and for ANG1 (endothelial cells in tissue). Please explain why normal control myometrium was used as positive controls for all 3 Abs.

7.     Figure 1. Why are there no images of Control tissue samples? Should include.

Minor concerns:

8.     Semiquantitative data, do the values represent intensity? Are the arbitrary numbers?

9.     Figure 2. Does the  Y-axis represent intensity? Please label.

10.  Page 9, Line 282. Unfamiliar with the term “benign cancerous growth?” can a benign tumor be a cancerous growth?

Author Response

Thank you for your comments and we hope that the corrections introduced are in line with your expectations.

Major concerns:

  1. Did the authors take into consideration the role of endogenous hormones on the expression of ANG1, CaSR, and FAK? Are the authors able to sort samples in premenopausal women by menstrual cycle phase (proliferative vs. secretory) to look at regulation of expression of these proteins? Does it change expression patterns?

Unfortunately, this is a retrospective and multicenter study, which results in the lack of access to the full medical history of all patients. Therefore, we cannot sort the preparations in the way suggested by the reviewer, although the results obtained in this way would undoubtedly be very interesting.

  1. No history of whether patients where on hormone replacement therapy, or birth control pills were taken or other exogenous forms of estrogen/progesterone at time of sampling and role these exogenous hormones could play on expression of proteins examined.

"The patients took no hormones for at least a month before the fibroid removal procedure." We have added this sentence to the manuscript.

  1. The activated/phosphorylated forms of these FAK and ANG1 should be assessed in tissues. These are signaling molecules (tyrosine kinases) and expression of total FAK or ANG1 gives no information on whether the protein is activated and has initiated downstream signaling pathways. These studies are critical studies needed for this manuscript.

Thank you for your valuable suggestion, but performing such studies requires the collection of new material and should be combined with the analysis of FAK and ANG1-dependent proteins, so we cannot add such results to this work. On the other hand, our goal was to demonstrate potential variation in expression in the fibroid and normal surrounding tissue.

  1. Please clarify tumor margin. Is it at the periphery of the tumor and consists of tumor tissue? OR is it non-tumor tissue at the periphery of the tumor? It is unclear as currently stated in the methods, although in the results it states, “tumor center, “tumor periphery.”

  1. Please describe in more detail the 14 tissue samples on each TMA. Were patient-matched paired tumor and tumor margin samples placed in the same block and stained under similar conditions? OR were all tumors in one TMA and tumor margin samples in another TMA. Authors only state clearly there are  2 non-tumor control samples per TMA. Please clarify.

For 4 and 5: In the description of the methods, we add a fragment: “The material was derived from uterine leiomyoma and muscle tissues outside the uterine myoma in its vicinity of the same patient.” Second, “The control group included tissue slices, morphologically unchanged, from 11 women without known fibroids upon histopathological examination. These women were operated on due to endometrial hyperplasia or genital prolapse. They were aged from 56 to 69 years, averaging 61. The women in the study and control groups had no additional comorbidities.”

  1. Why were non-tumor controls used as positive controls for the different antibodies (Abs)? The authors should have used positive control tissue that is known to highly express the proteins of interest. For example, caSR (human parathyroid or  kidney); for FAK (brain or lung) and for ANG1 (endothelial cells in tissue). Please explain why normal control myometrium was used as positive controls for all 3 Abs.
  2. Figure 1. Why are there no images of Control tissue samples? Should include.

For 6 and 7: The description in the figure has been changed, and images from controls have been prepared as a supplementary figure

Minor concerns:

  1. Semiquantitative data, do the values represent intensity? Are the arbitrary numbers?

the binding intensity of the antibody was measured as the brown surface area and this value is the basis for statistical calculations

  1. Figure 2. Does the  Y-axis represent intensity? Please label.

The chart caption has been changed accordingly

  1. Page 9, Line 282. Unfamiliar with the term “benign cancerous growth?” can a benign tumor be a cancerous growth?

It has been corrected

Kind regards

Mateusz de Mezer

Reviewer 2 Report

Comments and Suggestions for Authors

The scientific problem which is presented by the authors is still one the leading issues in current gynecology. Big number of our patients complain due to presence of myoma, often have not satisfactory pharmacological outcome as well time of diagnosis result in huge tumor findings…

The authors present quite interesting basic science approach to the fibroid evaluation. This study in my opinion should be treated as preliminary report due to the only one laboratory technique used for evaluation. Number of cases is clearly sufficient. Methodology is proper. References are up to date and properly picked.

I have some concerns and will point out some flaws which should be resolved prior decision of further steps of editorial process. However in regard to all below mentioned points after introduction explanations and clarifications I would vote for acceptance of publication revised paper.

1. The title should be revised -> The immunohistochemical expression; in the uterine fibroids…

2. References should be cited more precisely in the text, not like at the end of paragraph.

3. L.40 while we have precise classification of fibroids we should use it; palm-coein/FIGO.

4. Language and editorial improvement, l.45-47?; l.230-232? And other spots. L. 69. SPRMs instead of SPARMS…

5. l. 111 statement concerning FAK is not true. There are papers in this area, e.g. Chegini et al. PMID 125117589. To be deeply revised. 

6. l. 112-113 the aim of the study has to be reformulated as in its present form it is blurry what does it mean to evaluate importance?....

7. l. 123 vs. 127 it has to be clarified what exactly was the recommendation for the surgery as hypertrophy and hyperplasia are completely two different states... 

8. l. 123 genital is informative, however it is rather used POP term or full names... 

9. methodology section should be revised as there are missing data concerning suppliers, dilutions, reagents etc. 

10. What about other data? Only expression of 3 markers was evaluated? Any clinical or pathological data? Where exactly expression was observed in regard to cellular location? 

11. Table 1 is in general useless, especially in regard to two main columns - mean and range. I would suggest to combine data from table 1 and figure 2 to prepare column graphs with pointed p values for comparison purposes. 

12. What was the rationale for comparison of central and periphery of the tumor? How it was defined?

13. Descriptions on figure 1 are misleading - border/periphery. It all may be placed in panel with common description for rows and columns or just plain letters for every image separately. Control group images are missing. 

14. I would expect improvement of discussion in regard to expanded results, as well updated references. 

15. Conclusions should be reformulated. I would decrease strength of conclusion. Add observations on other tested markers. Speculate about directions of potential utility of tested proteins in diagnostic or therapeutic directions separately or as combined tests. 

Comments on the Quality of English Language

In general I don’t have any major concern, however it would be worth to double check the paper prior publication in regard to its language and editorial appropriateness. 

Author Response

Thank you for your comments and we hope that the corrections introduced are in line with your expectations.

  1. The title should be revised -> The immunohistochemical expression; in the uterine fibroids…

We propose a new title: “Analysis of expression of the ANG1, CaSR and FAK proteins in uterine fibroids”

  1. References should be cited more precisely in the text, not like at the end of paragraph.

We checked again and made individual shifts. We hope that we have achieved greater correctness and accuracy.

  1. 40 while we have precise classification of fibroids we should use it; palm-coein/FIGO.

In the section “Study Design” section, we explain that “FIGO classification was not used as the material originated from different clinics, but no pedunculated fibroid was found in the uterine cavity or the cervix. They were mainly FIGO grade 3 to 6 fibroids, according to Munro et al. 2011.”

  1. Language and editorial improvement, l.45-47?; l.230-232? And other spots. L. 69. SPRMs instead of SPARMS…

Thank you for your comments. Thanks to them, we have re-corrected the language by a native speaker

  1. 111 statement concerning FAK is not true. There are papers in this area, e.g. Chegini et al. PMID 125117589. To be deeply revised. 

In our opinion, although FAK1 is often studied in various types of endometrial cancers and is attributed to a role in regulating cell growth, the level of this protein in benign lesions such as uterine fibroids has not been investigated to date.

  1. 112-113 the aim of the study has to be reformulated as in its present form it is blurry what does it mean to evaluate importance?

The new version's aim description is: Of the various attempts at conservative treatment of uterine fibroids, not all of them have been accepted (e.g., the hormonal drug ulipristal acetate was withdrawn). The clinician and histopathologist believe that the deposition of calcium salts in the myoma cells is related to the inhibition of its growth (this is proven by the results of embolization - surgical procedure). This process involves Ang 1, which promotes vascular integrity and is involved in calcium signaling pathways. However, FAK, in addition to controlling the angiogenesis process through cell adhesion (including the fibroid), prevents its growth. Thus, the action of the three molecules we studied seems closely related. The study aimed to determine the importance of ANG1, CaSR, and FAK in uterine fibroids that could be applied in targeted therapy.

  1. 123 vs. 127 it has to be clarified what exactly was the recommendation for the surgery as hypertrophy and hyperplasia are completely two different states... 

It is described more preciselly

  1. 123 genital is informative, however it is rather used POP term or full names... 

It is corrected

  1. methodology section should be revised as there are missing data concerning suppliers, dilutions, reagents etc. 

The description of immunohistochemical staining has been completely corrected because, thanks to the reviewers' comments, it was noticed that by mistake the manual staining method was described using the Vector Laboratories system instead of the fully automatic system used in the case of the preparations described in this work, Bench-Mark ULTRA (Ventana Roche, Oro Valley, AZ, USA). This only editorial mistake results from the fact that in our laboratory we use both methods equally often.

  1. What about other data? Only expression of 3 markers was evaluated? Any clinical or pathological data? Where exactly expression was observed in regard to cellular location? 

Thank you for your valuable suggestion. Unfortunately, this is a retrospective and multicenter study, which results in the lack of access to the full medical history of all patients, so performing such tests requires collecting new material and should be combined with the analysis of FAK and ANG1-dependent proteins, so we cannot add such results to this work. On the other hand, we aimed to demonstrate potential differential expression in the fibroid and normal surrounding tissue. In turn, the cellular localization of the observed signals was consistent with that previously described, and for this reason, we did not describe it.

  1. Table 1 is in general useless, especially in regard to two main columns - mean and range. I would suggest to combine data from table 1 and figure 2 to prepare column graphs with pointed p values for comparison purposes. 

Thank you for your suggestion, but in our opinion, the proposed system of presenting the results, both in the form of a table and a figure, is more readable for clinical specialists potentially interested in seeking targeted therapy and for this reason we decided not to change it.

  1. What was the rationale for comparison of central and periphery of the tumor? How it was defined?

The material came from a leiomyoma uteris and  muscularis tissues outside the uterine fibroid in its vicinity.

  1. Descriptions on figure 1 are misleading - border/periphery. It all may be placed in panel with common description for rows and columns or just plain letters for every image separately. Control group images are missing. 

The description in the figure has been changed, and images from controls have been prepared as a supplementary figure

  1. I would expect improvement of discussion in regard to expanded results, as well updated references. 

  1. Conclusions should be reformulated. I would decrease strength of conclusion. Add observations on other tested markers. Speculate about directions of potential utility of tested proteins in diagnostic or therapeutic directions separately or as combined tests. 

For 14 and 15: because for the reasons described above it is not possible to extend the results, we also did not conduct the discussion

Kind regards

Mateusz de Mezer

Round 2

Reviewer 1 Report

Comments and Suggestions for Authors

Many of the major criticisms of the manuscript concerning the assessing of activation  status of receptors were not addressed by the reviewers, which is critical. Total protein expression does not indicate whether the protein is involved in the signaling cascade which is needed to elicit biological effects. There are commercially available phospho-antibodies that can be used in FFPE samples and IHC.